# Non-Random Distribution of *Boreus hyemalis* Among Bryophyte Hosts: Evidence from Field and Laboratory Tests

**DOI:** 10.3390/insects15110878

**Published:** 2024-11-08

**Authors:** Petr Pyszko, Michaela Drgová, Vítězslav Plášek, Pavel Drozd

**Affiliations:** 1Department of Biology and Ecology, Faculty of Science, University of Ostrava, Chittussiho 10, 710 00 Ostrava, Czech Republic; petr.pyszko@osu.cz (P.P.); michaela.drgova@osu.cz (M.D.); vitezslav.plasek@osu.cz (V.P.); 2Institute of Biology, University of Opole, Oleska 22, 45-052 Opole, Poland

**Keywords:** boreidae, bryophagy, bryophytes, insect–plant interactions, non-random distribution

## Abstract

This study explores whether the winter-active insect, *Boreus hyemalis*, shows a non-random distribution among different moss species in its natural habitats and in the laboratory. In field surveys, *B. hyemalis* was found on 21 moss species, but showed a particular association with *Dicranella heteromalla* and *Hypnum cupressiforme*. In laboratory tests, *Polytrichum formosum* was the most frequently selected species. Although some discrepancies between field and lab results were noted, our findings suggest that *B. hyemalis* does not associate with mosses randomly, but tends to select certain species. This research helps us understand how these insects interact with their mossy environments.

## 1. Introduction

Host specificity among herbivorous insects has become a central theme in ecological research due to its implications for biodiversity and ecosystem functioning [1,2]. Recent studies have intensified efforts to understand the variability in host plant selection, which is influenced by a multitude of ecological and evolutionary factors [3]. However, plant species consistently exhibit variability in the richness of herbivorous insect species they support [4]. This variability can result from differences in the abundance of predators and parasitoids [5], but it is primarily driven by host-related factors such as the number of young leaves, leaf production, palatability, water content, plant height, nitrogen content, and phenotypic variability [6,7,8]. Host plant preference differences have been well-documented for numerous herbivores on herbaceous and woody plants, annuals and perennials, and even aquatic species [9]. However, with few exceptions, bryophytes have largely been overlooked. Consequently, while data exist for certain groups of herbivorous insects, the host specificity of many other taxa or guilds, including nearly all bryophagous insects, remains poorly understood.

Unlike vascular plants, bryophytes as host plants have received relatively little attention in ecological research. While bryophytes serve as food for a limited number of insect species, they provide shelter with favorable physical conditions and protection from predators for a wide variety of insects [10]. The preference for mosses as a microhabitat can even influence the dietary choices of bryophagous insects [11]. Several insect taxa are currently presumed to be bryophagous, including certain members of Ephemeroptera [12], Thysanoptera [13], Lepidoptera [14], Auchenorrhyncha (Hemiptera: [15]), Coleorrhyncha (Hemiptera: [16]), Tipulidae (Diptera: [11]), Tetrigidae (Orthoptera: [17]), and Byrrhidae (Coleoptera: e.g., [18]). The ecological characteristics of these insects, including their host specificity, preferences, and adaptations to bryophagy, remain largely unexplored.

Most associations between bryophagous species (or higher taxa) and their host bryophytes are inferred from the presence of bryophages on specific bryophytes [19,20,21] and only occasionally from direct identification of bryophytes in their digestive tracts [22,23,24]. However, few studies have compared these associations with the actual abundance of bryophytes in the field (e.g., [17]). Controlled experiments providing strong evidence of host associations are also rare, including no-choice assays [25] or comparisons of one moss species with several vascular plants [26]. Generally, experiments involving multiple-choice tests of preferences among several bryophyte species are lacking, with a few exceptions, such as the larval performance comparison for *Tipula montana* Curtis, 1834 across five moss species [11] and the simultaneous testing of several host species assessed by acceptance rank on a five-degree scale for *Caurinus dectes* Russell, 1979 [27], a boreid feeding primarily in stem-mines or galleries of epiphytic liverworts [27,28,29].

The Boreidae family is among the most intriguing groups of bryophages, known for their peculiar flightless appearance, unusual winter activity, and closer relationship to parasitic fleas than to other mecopterans [30]. However, little is known about the host preferences of boreids, particularly those of the genus *Boreus*. These insects are rarely recorded in standard surveys, often eluding the attention of ecologists due to their obscure phenology, winter occurrence, and highly clumped distribution. Thus, the diet of *Boreus* species has been subject of long debate, with opinions ranging from obligatory zoophagy for both larvae and adults [31,32,33,34,35] to a transition from larval herbivory to adult zoophagy, potentially combined with facultative bryophagy [35,36,37]. However, multiple studies have firmly established *Boreus* as primarily phytophagous, with larvae and adults feeding on bryophytes [27,38,39,40,41,42,43,44,45,46,47]. This conclusion is supported by their short and stout mandibles with subapical teeth suited for feeding on bryophytes [48]. Withycombe [35] was the first to note that both larval and adult stages feed on young shoots and decayed phylloids, and that larvae also feed on rhizoids of mosses and liverworts [27,29]. Gathering information on the preferences of *Boreus* species is still particularly challenging since most findings occur in winter, where they are more often found on snow rather than directly on bryophytes.

We focused on *Boreus hyemalis* (Linnaeus, 1767) (Boreidae: Mecoptera, Figure 1), a relatively abundant species in central Europe, typically found in heaths, moors [49,50], and continuous forests dominated by beech *Fagus sylvatica* L. or hornbeam *Carpinus betulus* L. [38,51]. *B. hyemalis* is believed to primarily inhabit these environments as they are rich in mosses such as *Hypnum cupressiforme* Hedw., *Polytrichum formosum* Hedw., or *Dicranum scoparium* Hedw. [31,52]. To date, *B. hyemalis* has been associated with a variety of moss species, including *Atrichum undulatum* (Hedw.) P. Beauv., *Brachythecium rutabulum* (Hedw.) B. S. G., *B. salebrosum* (Hoffm. ex F. Weber et D. Mohr) Schimp., *B. velutinum* (Hedw.) Schimp., *Campylopus gracilis* (Mitt.), A Jaeger, *Dicranum montanum* Hedw., *D. scoparium*, *H. cupressiforme*, *Plagiothecium curvifolium* Schlieph. ex Limpr., *P. formosum*, and *P. commune* Hedw. [46,49,50,51,53]. *B. hyemalis* larvae have also been found among the roots of *Mnium hornum* Hedw. [54], and, according to Fraser [46], oviposition preferably takes place on *P. commune*, although other species of bryophytes are also likely used [49]. The species is active mainly in the winter months, from October to February, with peak activity observed in December and January. It is highly adapted to survive and remain active at low temperatures, making it one of the few insects that thrive during winter [55].

Except for the aforementioned study on *C. dectes*, the “association” with bryophytes typically refers to records of bryophyte species from which boreids have been collected or observed, which also applies to species of the genus *Boreus,* including *B. hyemalis* species. However, these observations have not been quantified by comparing them with the actual abundance of host plants in the field. This gap highlights the need for a study that quantifies the association between *B. hyemalis* and selected bryophyte species, determining whether these mosses are occupied by boreids more than their field abundance would suggest.

Our objectives were threefold: (i) to test whether the presence of *B. hyemalis* on different moss species in the field corresponds to the proportional coverage of these mosses (i.e., whether their distribution is random or indicates specific preferences); (ii) to assess whether boreid presence at a site with higher boreid abundance is influenced by the availability of certain moss species or whether other environmental factors play a role; (iii) to test selection tendencies of *B. hyemalis* among selected moss species under laboratory conditions, focusing on potential differences between sexes and responses to different morphotypes of *H. cupressiforme*. The inclusion of *H. cupressiforme* in the study was based on its morphological diversity and its high abundance in the field, not on any prior assumption regarding its preference by the species.

## 2. Materials and Methods

### 2.1. Field Tests

The first phase involved assessing whether the selection of mosses where boreids are found is random or exhibits selection tendencies. During the winter of 2020/2021 (from 27 November to 6 March), we visited 10 sites in the Czech Republic with suitable habitats and under favorable weather conditions as described in the literature—sunny, not windy, with temperatures around or slightly above 0 °C [38,39,56]. We defined each site as a specific locality covering an area of an irregular shape, ranging from approximately 50 × 50 to 100 × 100 m, characterized by homogeneous ecological conditions and selected based on known occurrences of *B. hyemalis* from previous records, accessibility during winter months, and representation of typical habitats where the species is found, such as forests dominated by beech or hornbeam [38,51]. The irregular shape was caused by natural boundaries, such as forest edges or other natural obstacles. Each site was rich in bryophyte diversity, providing ample microhabitats for *B. hyemalis*. A map illustrating the geographical distribution and specific locations of these sites is provided in Figure 2. We aimed to locate at least 10 boreid individuals per site directly on mosses, avoiding those found on snow. Six sites yielded more than 10 individuals: Budišovice (49°51.95′ N, 18°3.09′ E, *n* = 12), Darkovičky (49°55.43′ N, 18°10.81′ E, *n* = 13), Děhylov (49°52.55′ N, 18°9.91′ E, *n* = 15), Pustá Polom (49°50.30′ N, 17°59.72′ E, *n* = 10), Skalka/Bystřice (49°36.60′ N, 18°41.72′ E, *n* = 11), and Smolkov (49°53.73′ N, 18°4.45′ E, *n* = 11).

For each boreid individual found, we recorded the moss species directly associated with it. If the individual was found within approximately 15 cm of multiple moss species, we estimated the percentage cover of each moss species within that area, following Pyszko et al. [57]. The association of the boreid was then proportionally divided among these moss species based on their relative cover. For example, if two moss species were present within the 15 cm radius, covering 40% and 60% of the area, respectively, the boreid individual was assigned as 0.4 to the first species and 0.6 to the second. This proportional assignment allowed us to compare the moss composition directly around boreids with the overall moss composition at the site.

Additionally, at each site, 10 randomly placed 1 × 1 m quadrats were established. These quadrats were randomly placed throughout the entire locality where boreid individuals were found to occur, ensuring coverage of all potentially suitable microhabitats for boreids. We avoided placing quadrats in obviously unsuitable habitats (e.g., grassy meadows outside the forested area) to focus on areas relevant to boreid ecology. Within the suitable habitat, quadrats were placed independently of the specific locations where boreids were observed, including areas where boreids were not found during our surveys. This approach allowed us to capture variability within the habitat, recognizing that boreid counts of zero can result from either low preference for certain microhabitats or from stochastic factors, rather than from absolute habitat unsuitability. Within each quadrat, all bryophyte species were identified, and their cover was estimated using the Braun-Blanquet scale. For class 2 (which represents species cover between 5% and 25%), we further subdivided this class into three categories: 2 m (5–10%), 2a (10–15%), and 2b (15–25%), to capture finer variations in bryophyte cover. This modification allowed for a more precise estimation of moss cover, particularly for species that occurred in intermediate abundances [58]. These cover estimates were then converted into approximate moss cover values for the quadrats, and the average cover of each moss species at the site was calculated. For subsequent analyses, we focused on the 10 most prevalent moss species (those present at least at 4 out of 6 sites with an average cover > 2%), while other species were grouped into a single category labeled as “rare species”.

On 3 December 2021, we conducted a detailed survey of the microhabitat requirements of boreids at the Darkovičky site. This location provided suitable conditions for the research, being part of a *fagetum nudum*, a nearly monocultural beech forest (*F. sylvatica*) characterized by a very sparse or absent shrub and herb layer, due to natural leaf litter, which inhibits ground vegetation growth. The area exhibited variable understory shading, creating diverse light conditions. Despite limited diversity in ground vegetation, the site supported a rich moss layer with a relatively simple species composition dominated by 10 species: *Dicranella heteromalla* (Hedw.) Schimp., *A. undulatum*, *Pohlia nutans* (Hedw.) Lindb., *Aulacomnium androgynum* (Hedw.) Schwägr., *Fissidens bryoides* Hedw., *H. cupressiforme*, *B. rutabulum*, *B. salebrosum*, *M. hornum*, and *Plagiothecium undulatum* (Hedw.) Schimp. These mosses formed distinct cushions, adequately spaced apart. This site had previously yielded a sufficient number of boreids during the prior winter, suggesting that a similar abundance would be present in the current winter.

We systematically established 50 quadrats, each measuring 0.5 × 0.5 m^2^, ensuring that each quadrat contained one moss cushion. Each quadrat was photographed before being thoroughly searched for boreid presence, noting any individuals found, their abundance, and sex. In total, 58 individuals were recorded. For each quadrat, we used ImageJ software (version 1.53t) [59] to analyze the total area of the quadrat covered by moss and the light conditions. We categorized the degree of association between the moss cushion and the nearest tree trunk into five levels, ranging from 0 (freely located in open space) to 1 (moss cushion directly on the trunk). Additionally, we assessed the relative area occupied by each of the ten moss species in the photograph and determined the moss species richness within each cushion.

### 2.2. Laboratory Test

We collected *B. hyemalis* (*n* = 30) in February 2017 from the snow surface in the Beskydy Mountains, Czech Republic: Okrouhlice (49°31.51′ N, 18°34.82′ E), Skalka/Horní Lomná (49°32.45′ N, 18°34.81′ E), Skalka/Bystřice (49°36.28′ N, 18°40.91′ E). The sex of each individual was recorded to be used as an explanatory variable in the analyses. Since the individuals were collected from the field, it was not possible to determine their exact age or mating experience. However, boreid individuals survived in captivity for an average of 12.75 ± 1.191 days (including four individuals that died during acclimation before the experiment), suggesting they were in relatively good health upon collection in context of their standard survival in captivity [29,44,45,59]. The individuals were maintained separately in plastic boxes within a growth chamber set to an 8:16 h light cycle, at 8 °C, and 75% humidity. The temperature of 8 °C was chosen as a compromise between the need to keep the individuals alive as long as possible (since lower temperatures prolong their lifespan) and the technical limitations of the growth chamber, when they were kept, which could not sustain temperatures below 8 °C for extended periods. After a 5-day acclimation period, the insects were randomly assigned to a feeding experiment (*n* = 28).

Each boreid was tested separately in its own circular plastic box. This ensured that each individual’s behavior was not influenced by the presence of other boreids. The plastic boxes used for the tests measured 25 cm in diameter and 9 cm in height, providing ample space for the insects to move and choose among the moss species. The walls of the boxes were smooth to prevent the boreids from escaping, and the lids were perforated to ensure proper ventilation. The base was lined with moistened cotton wool to maintain humidity and prevent desiccation during the experiment. The mosses were randomly arranged along the walls of the plastic boxes, ensuring that each species had an equal chance of being selected by the boreids during the tests. The mosses used for the test were standardized to approximately 5 cm^3^ pieces, ensuring consistency in size across trials. After two hours, we observed the position of the individual for 120 s and recorded the moss species on which it was found. If a boreid visited more than one moss during the observation period, its choice was divided among the mosses based on the length of stay. Between measurements, each individual was starved for at least two hours. Each individual underwent up to 16 repeated trials, with sufficient intervals between trials to minimize carry-over effects; individuals who died after only a few repetitions were excluded from the final dataset, resulting in 22 individuals being used in the final analysis with up to 16 repeated measurements each.

We tested 14 different mosses, including 11 distinct species, which varied in morphology from compact, small forms to highly branched, tall forms: *B. rutabulum*, *B. salebrosum*, *Bryum argenteum* Hedw., *Ceratodon purpureus* (Hedw.) Brid., *D. scoparium*, *P. curvifolium*, *P. undulatum*, *Pleurozium schreberi* (Brid.) Mitt., *P. nutans*, *P. formosum*, and *H. cupressiforme*. *H. cupressiforme* was tested due to its well-known morphological variability, which is greater than that of most other moss species, in four of its most common morphological variants: filiforme, reticulate, turgid, and the usual form. The high number of morphologically distinct forms of *H. cupressiforme* is sometimes reflected in its taxonomy [60] and this significant variability has been the subject of previous studies, including our own prior work [61]. Among the selected mosses, we also included species that were not found in environments inhabited by boreids (*B. argenteum*, *C. purpureus*, *D. scoparium*, *P. schreberi*), to test if these mosses would be less selected by the insects. All mosses were sampled in January 2017 and acclimated in growth chambers for three weeks before the feeding experiments.

The four morphotypes of *H. cupressiforme* used in the laboratory tests were defined based on distinct morphological characteristics, including the size of moss cushions; the length, width, and branching patterns of the stems; and the spaces between the stems: the *turgid* morphotype is characterized by robust mats consisting of stems 5–8 cm long and branches up to 3 mm wide. The free space among individual stems in the mats ranges from 0.3 to 1.5 cm. This morphotype typically grows on forest floors and in forest litter, and does not occur as an epiphyte on tree bark. The *usual* morphotype corresponds to the description in most identification keys and is the most common morphotype for this species. It is characterized by medium-sized mats with stems 3–5 cm long and branches up to 2 mm wide. The free space among individual stems in the mats is 0.2–1.0 cm. This morphotype grows on a variety of substrates, including forest floors, stones, and rock walls, and it can also occur as an epiphyte on both deciduous and coniferous tree bark. The *filiforme* morphotype is characterized by small, slender mats consisting of non-branching (or very rarely branching) stems up to 5 cm in length and up to 1 mm in width. The free space among individual stems in the mats is 0.4–0.6 cm. This morphotype grows mostly on stones, boulders, and rock walls, and it also occurs as an epiphyte on tree bark. It is usually confined to vertical surfaces. The *reticulate* morphotype is similar to the filiforme morphotype but differs by having slender, richly and distinctly branched stems up to 5 cm long and branches up to 1 mm wide. The free space among individual stems in the mats is 0.3–0.6 cm. This morphotype also grows on stones, boulders, and rock walls, and occurs as an epiphyte on tree bark, being predominantly confined to vertical surfaces [61]. These morphotypes were chosen to capture the morphological variability of *H. cupressiforme* in different habitats and to assess whether these structural differences influence boreid selection tendencies.

### 2.3. Data Analysis

We analyzed and visualized the data using R version 4.3.3 [62]. For the initial field data (to determine whether the distribution among mosses is random), we performed an ANCOVA using a generalized linear mixed model (GLMM) from the “lme4” package (v1.1.35.3) [63], with a negative binomial distribution from the “MASS” package (v7.3.60.0.1) [64]. The response variable was the number of boreids captured on each moss, with moss cover as a covariate to account for differences in moss abundance and to identify potential discrepancies between moss cover and boreid presence, expecting that, under random distribution, the relationship between the number of boreids and moss cover would be approximately linear. Thus, in this analysis, the expected random distribution of boreids serves as a control, and deviations from this distribution indicate their selection tendencies for certain moss species. The explanatory variables were moss species and the interaction between moss species and sex. The specific site was included as a random term. The significance of the results was tested by comparing the Akaike information criterion (AIC) with the corresponding simplified model.

For the additional field data, which aimed to identify factors influencing boreid presence at the microhabitat level, we applied a weighted generalized linear model with a binomial distribution and logit link function. The response variable was the presence or absence of boreids in each quadrat, with the weights assigned based on boreid abundance or set to 1 if absent. Explanatory variables, selected through stepwise selection based on AIC, included the proportional representation of the 10 present moss species, total moss cushion area within each quadrat, light conditions, degree of association of the quadrat with a tree, and overall moss richness.

For the laboratory selection tendency testing data, we applied a GLMM with a binomial distribution, where individual boreids were treated as a random term to account for repeated measurements from the same individual. Each boreid was observed up to 16 times in separate trials, and the repeated measure’s designs were incorporated into the model to prevent pseudoreplication and ensure robust statistical inference. This approach allowed us to consider the within-individual correlations arising from multiple observations of the same individual. To avoid potential positional bias, the order of moss species in each trial was randomly changed, allowing us to capture unbiased selection tendency patterns. The response variable was the presence or absence of an individual on a given moss species. Potential explanatory variables included moss species, the interaction between moss species and sex, the order of individual trials (to test for possible learning effects in selecting certain moss species), and the interaction between trial order and moss species. Only significant variables were retained in the final model, with the model’s significance tested by comparison with the appropriately simplified model.

To assess whether the observed ordering of the four selected mosses, (i) those not commonly found in boreid habitats or (ii) those forming small, compact cushions, deviates from a random arrangement, we conducted a permutation test. Specifically, we generated 10,000 simulations in which the ranks of the four mosses were randomly assigned within the 14 available positions (representing 14 species or morphological types of mosses). For each simulation, we determined whether the randomly assigned ranks were as unfavorable as or more unfavorable than the observed rank. The *p*-value was calculated as the proportion of simulations where the simulated rank was equal to or worse than the observed rank. This post hoc analysis was performed to support our discussion by assessing the likelihood that the observed moss rankings occurred by chance. All graphs were created using the “jtools” (v2.3.0) [65] and “ggplot2” (v3.5.1) [66] packages. All figures depicting model estimates also include raw data points representing individual observations to provide a comprehensive view of the observed data alongside the statistical model predictions. Raw data for the field selection test, microhabitat test, and lab selection test are available in the Appendix A.

## 3. Results

### 3.1. Field Tests

Across the six sites, we found a total of 72 boreids on 21 different moss species. The highest number of individuals was found on *D. heteromalla* (*n* = 21), followed by *H. cupressiforme* (*n* = 15), and *A. undulatum* (*n* = 8), which were also among the most abundant mosses in the field. The number of boreids found on each moss species was positively correlated with moss cover (χ^2^_62,1_ = 11.08, df = 1, *p* < 0.001). Beyond this, the number of boreids varied significantly among moss species (χ^2^_62,10_ = 25.76, *p* = 0.004), with *D. heteromalla* and *H. cupressiforme* being selected (Figure 3) more often than would be expected based on their field representation. The sex of the boreids did not influence their moss selection tendencies (*p* > 0.050).

The survey of microhabitat factors influencing boreids revealed that light conditions, the degree of quadrat association with a tree, and, surprisingly, even moss cover within the quadrat (i.e., the size of the moss cushion) did not play a significant role (*p* > 0.050). Of all the moss species, the presence of boreids was positively associated with only one, *H. cupressiforme* (χ^2^_48,1_ = 8.44, *p* = 0.004, Figure 4a). Additionally, the overall moss richness was a significant factor, with the probability of boreid presence increasing as moss richness increased (χ^2^_47,1_ = 28.89, *p* < 0.001, Figure 4b).

### 3.2. Laboratory Test

A total of 22 *B. hyemalis* individuals were included in the final laboratory test analysis. The laboratory test revealed significant tendencies in selection among different moss species (χ^2^_2550,13_ = 28.56, *p* = 0.008, Figure 5), with *P. formosum* being the most frequently selected species. Among the other mosses, the most commonly selected were two variants of *H. cupressiforme* (filiforme and reticulate morphotype), as well as *B. salebrosum*. The least selected was turgid morphotype of *H. cupressiforme*. *P. formosum* was the only species that differed significantly from the others, while the remaining mosses did not show significant differences from each other in pair-wise comparisons. During the observation period, no significant shifts in selection tendencies were detected, and there were no significant differences between sexes (both *p* > 0.050), although a slightly higher selection for *P. formosum* was observed in males.

The observed ranking of the four moss species atypical for boreid habitats or never observed in their vicinity was 6th, 11th, 12th, and 13th place. The permutation test which compared these rankings against 10,000 randomly generated arrangements within 14 possible positions indicated that this rank was unlikely to have occurred by chance (*p* = 0.019). Similarly, the observed ranking of the four mosses growing in low, compact cushions was 7th, 10th, 11th, and 12th place, which the permutation test also determined to be non-random (*p* = 0.034). These results suggest that the selected moss species were significantly less-chosen than would be expected under a random distribution.

## 4. Discussion

Overall, we found boreids on 21 different moss species, which is quite a lot considering the 72 individual boreids observed. It is challenging to determine why these specific mosses were selected, as our objective was not to investigate their actual feeding preferences, but rather to identify the host plants where they are more likely to be found than would be expected based on the host’s environmental abundance. However, if the boreids were indeed consuming these mosses, this would be consistent with studies showing that more than 40 species of mosses from nine orders and 19 families can serve as potential hosts for different *Boreus* species [27,40,41,42,48,49,53,67,68,69,70]. Similarly, *Caurinus dectes*, for which host preferences are the most studied, frequently feeds on 14 species of liverworts, but when the primary choice is unavailable, it consumes an additional 18 species of liverworts or mosses [27,29,49]. Furthermore, *Hesperoboreus notoperates* Cooper, 1972, the second most studied boreid, has been observed on at least 16 species of mosses from the families Grimmiaceae, Orthotrichaceae, Pottiaceae, and Thuidiaceae [40,41].

Despite being found on 21 different moss species, boreids were frequently observed on *Dicranella heteromalla* and *Hypnum cupressiforme*, with a total of 36 individuals, accounting for half of the observed boreids. These are also typically the most abundant mosses in the types of habitats where we searched for boreids. Both mosses are commonly found in woodland habitats, thriving in shaded, moist conditions, often growing on the forest floor, decaying wood, or at the bases of trees. They prefer temperate deciduous or mixed forests, particularly beech forests [71], which are also suitable habitats for boreids, as they are found in continuous forests dominated by beech (*Fagus sylvatica*) or hornbeam (*Carpinus betulus* L.) [38,51]. Nevertheless, we observe a selection for these mosses that exceeds their field abundance. This aligns with the findings of [55], who suggested that moss species composition is crucial for boreids, which should occur only on a limited number of moss species, slightly questioning the broader range of host plants mentioned earlier. Some studies suggest that herbivorous species may prefer the most abundant host plants, which could be advantageous in terms of foraging efficiency. The most common effect of frequency-dependent host choice is to increase encounter rates and/or preference for the most common host or prey type [72]. For example, the mosses that serve as food for *H. notoperates* have all been noted [40] as common species, and this generalization may also apply to *Boreus* [71].

In our study of microhabitat preferences, we aimed to test the influence of light, the degree of moss association with trees, and the presence of various dominant moss species in environments with high boreid populations. Despite temperature being a more critical factor than light in adult boreid activity [43], light conditions might also play a role. Furthermore, we were interested in the impact of moss association with trees, as *B. hyemalis* should inhabit mosses on large trees, stumps, and trunks. Surprisingly, neither factor appeared to have a significant effect. On the other hand, there was again a selection tendency for *H. cupressiforme* and a higher likelihood of boreid presence in moss cushions with greater moss species richness. Species richness is often positively associated with herbivore damage [73], especially in polyphagous species. Mosses contain high levels of repulsive or digestion-hindering secondary compounds [74], which may support the dietary-mixing hypothesis—the interplay between nutrient balancing and toxin dilution in foraging by a generalist insect herbivore [75]. Active mixing of multiple moss species in the diet, where greater richness is even observed in the gut than in the immediate environment, has been demonstrated for beetles of the family Byrrhidae [57].

We conducted selection tendency experiments for *B. hyemalis* based on choices among a relatively large number of potential moss host species. To a certain extent, the boreids were found on all offered mosses, although there were differences among them. The rank-plot of selection tendencies resembles a broken-stick null model describing random niche apportionment, which suggests either a possible wide-range polyphagy, as discussed above, or simply a near-random distribution of boreids among mosses in the laboratory. This may be due to the small size of the moss samples used under unnatural laboratory conditions, which did not lead to strong tendencies toward selection. Nevertheless, some interesting trends were observed.

According to Penny [44], mosses that grow in low, compact cushions, whose phylloids and especially rhizoids are tightly matted, such as those in the orders Grimmiales or Isobryales, may harbor more boreids than those growing in coarse, open, loose clumps. One possible reason may be the greater predation pressure from carabid beetles in the latter bryophytes, but these mosses may also lose moisture quickly and generally lack a suitably fine rhizoid mat, which is likely essential as a food source and microhabitat, especially for boreid larvae [40,76]. Additionally, larval and pupal *H. notoperates* are found abundantly in compact mosses [40]. This contrasts with our results, as the most frequently selected moss, *Polytrichum formosum*, is a species with typically high stems and an open structure. The next four frequently chosen mosses were pleurocarpous species with loose cushions, and the highest-ranked moss with approximately low, compact cushions was *Dicranum scoparium*. Moreover, mosses with the lowest and most compact cushions among those tested, namely, *Ceratodon purpureus*, *Pohlia nutans*, and *Bryum argenteum*, were ranked 10th, 11th, and 12th for selection, respectively, which overall indicates a significant tendency to avoid mosses with low, compact cushions. The tendency to avoid compact mosses may be related more to the needs of the difficult-to-study larvae rather than adults. For some boreids, a known preference shift occurs from the larval to adult stages [77].

The observed avoidance of mosses with low, compact cushions in the laboratory test sharply contrasts with the fact that a large portion of individuals in the field were found on *D. heteromalla*. The absence of *D. heteromalla* in the laboratory test is a limitation that prevents a direct comparison between its field and laboratory performance. Conversely, in the laboratory, there was a strong preference for *P. formosum*, which showed low capture rates in the field. This highlights the need for future studies to include all relevant moss species observed in the field to enable a more comprehensive comparison between field and laboratory preferences. The discrepancy suggests that *P. formosum* might actually be a frequently selected moss in the field as well, but its tall growth and complex structure may make it more difficult for individuals to be located. Furthermore, boreids exhibited slight (though statistically not significant) sex-related variance in moss selection tendencies, with males showing a higher selection tendency for *P. formosum* than females. This may be related to the premating behavior of boreids. Males often position themselves in waiting stances on moss stems, where they can spot a female and approach her unexpectedly while she is feeding [40,78]. Therefore, mosses with tall stems and structures that impede the view of feeding females may offer males more mating opportunities compared to mosses with low and compact cushions.

Even closely related bryophyte species could be selected differently by boreids, as shown by differing reactions to similar species in feeding trials with *C. dectes* [27]. Similarly, we observed different selection tendencies for morphological types of the same species, *H. cupressiforme*, although these differences were not statistically significant. The filiforme morphotype was the second most frequently selected offer, whereas the turgid morphotype was the least selected. These variations occurred even though all morphotypes were sampled (a) from the same sites to minimize variance between types, albeit with possible spatial variance between them, (b) during winter to exclude the effect of any potential anti-herbivorous defense induced by other bryophagous insects, and (c) sampled populations were acclimated in growth chambers to reduce physiological variability. The frequently selected filiforme morphotype is characterized by slender, non-branching stems with free space among individual stems, whereas the least selected turgid morphotype is characterized by robust, densely spaced stems. These features, particularly the space among stems, may play a role in the selection tendencies of different populations by bryophagous insects, which may perceive these morphological types as completely different species, some of which may be selected while others may be neglected. The differences in boreid selection tendencies toward different morphotypes of the same moss may be explained not only by morphological differences but also by chemical and physical properties, as it has been shown that beetles of the family Byrrhidae can distinguish morphotypes even when differences in morphological structure are suppressed [61].

In our test, we also included mosses that are not typically found in environments with boreids. *Pleurozium schreberi* is commonly found in boreal forests but also in dry grasslands [79]. *B. argenteum* thrives in highly disturbed, sun-exposed environments, growing on sidewalks, walls, and open ground, and *C. purpureus* tolerates a wide range of environments but commonly grows in dry, sunny places, including bare soil or rooftops. It can also colonize burnt areas and open spaces in cities. Both of these cosmopolitan species exhibit high desiccation tolerance [80]. Only *Plagiothecium curvifolium* prefers moist, shaded environments and is often found in deciduous forests, including beech forests [81], which could be considered suitable for boreids, yet we have not found this species in boreid habitats. The fact that these mosses ranked statistically significantly among the least selected may not reflect intentional selection tendencies by boreids but rather a correlation with environmental availability and suitability. This suggests that the observed patterns in boreid moss associations could be influenced by habitat factors. Therefore, these findings should be interpreted cautiously and require further validation through targeted experiments. Despite the apparent contradiction between the avoidance of certain mosses and the observed wide host range of *B. hyemalis*, these behaviors are not mutually exclusive. While boreids may utilize a diverse array of familiar moss species to maintain a balanced diet and dilute the effects of potential toxins, they could simultaneously avoid unfamiliar mosses due to the risk of unknown toxicity or lack of nutritional value. This pattern aligns with the dietary mixing hypothesis, which suggests that generalist herbivores aim to maximize dietary diversity to counteract negative effects of plant secondary metabolites, but remain cautious of new, untested food sources [82].

Our study employed three distinct experimental approaches—field surveys, microhabitat analysis, and laboratory tests—each offering unique insights while having specific limitations. The field surveys allowed us to observe *B. hyemalis* in its natural environment; however, the varying abundance of bryophyte species across sites may have influenced perceived selection tendencies. The microhabitat analysis provided detailed data on the influence of specific environmental factors, but the complexity of natural habitats made it challenging to isolate the effects of individual variables. Conversely, the laboratory tests were conducted under controlled conditions to minimize external influences; however, the artificial setup and relatively small sample sizes of boreids (though substantial, given the difficulty in obtaining these organisms and the careful repeated-measures design, providing robust data) and the small sample size of mosses may not fully reflect natural selection tendencies. These inherent differences in environmental conditions, as well as the variability in available bryophytes across experiments, likely contributed to the inconsistencies observed among the three approaches and should be considered when interpreting the results. Additionally, the incubation temperature of 8 °C is higher than the optimal conditions for *B. hyemalis* in the wild. While this may have accelerated their metabolism and potentially increased hunger levels, all individuals were subjected to the same conditions, minimizing any bias in comparative results among different moss species. Furthermore, since all individuals were kept separately throughout the study to avoid pseudoreplication, mating behavior and reproduction were not factors influencing the outcomes.

## 5. Conclusions

Our study provides new insights into the host selection tendencies of *Boreus hyemalis*, particularly highlighting its frequent occurrence on certain moss species, especially *Hypnum cupressiforme*, which exceeds its field abundance. However, our findings also revealed inconsistencies between field observations and laboratory results. While *H. cupressiforme* showed a tendency to be selected more frequently in the field, it did not exhibit a statistically significant tendency toward selection in the laboratory experiments. This discrepancy suggests that environmental context plays a crucial role in shaping boreid selection tendencies and highlights the importance of interpreting laboratory results with caution. The observed patterns in both field and laboratory conditions suggest that microhabitat factors such as moss species richness and specific morphological traits of moss cushions may play a significant role in the distribution and of boreids. These findings contribute to our understanding of the ecological requirements of this cryptic species.

## Figures and Tables

**Figure 1 insects-15-00878-f001:**
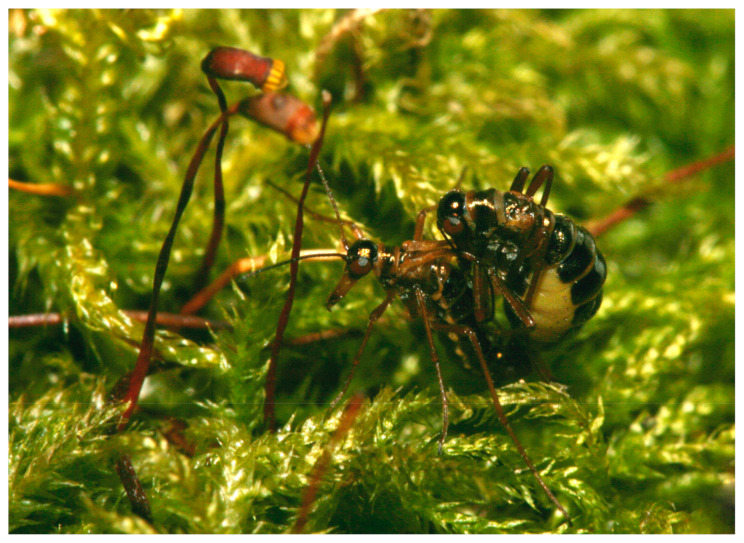
*Boreus hyemalis* (Linnaeus, 1767), a rare, winter-active insect, perched on a moss cushion in its natural habitat. The image shows mating individuals. The typical position where the male is located at the bottom and the female at the top. This species is primarily associated with mosses and is most often found in forested environments dominated by beech and hornbeam trees (author Jan Ševčík).

**Figure 2 insects-15-00878-f002:**
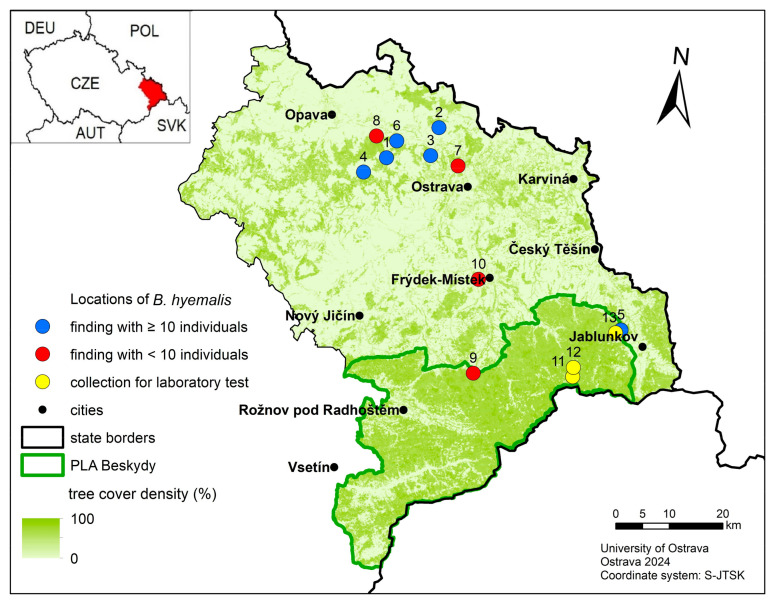
Map of study sites in the Czech Republic. Blue markers represent the six sites that yielded sufficient numbers of *Boreus hyemalis* (Linnaeus, 1767) individuals for the field study (1. Budišovice, 2. Darkovičky, 3. Děhylov, 4. Pustá Polom, 5. Skalka/Bystřice, 6. Smolkov). Red markers indicate sites that were visited but yielded low numbers of boreids (7. Lhotka, 8. Mokré Lazce, 9. Čeladná, 10. Lysůvky), while yellow markers represent sites where boreids were collected for laboratory tests (11. Okrouhlice, 12. Skalka/Horní Lomná, 13. Skalka/Bystřice).

**Figure 3 insects-15-00878-f003:**
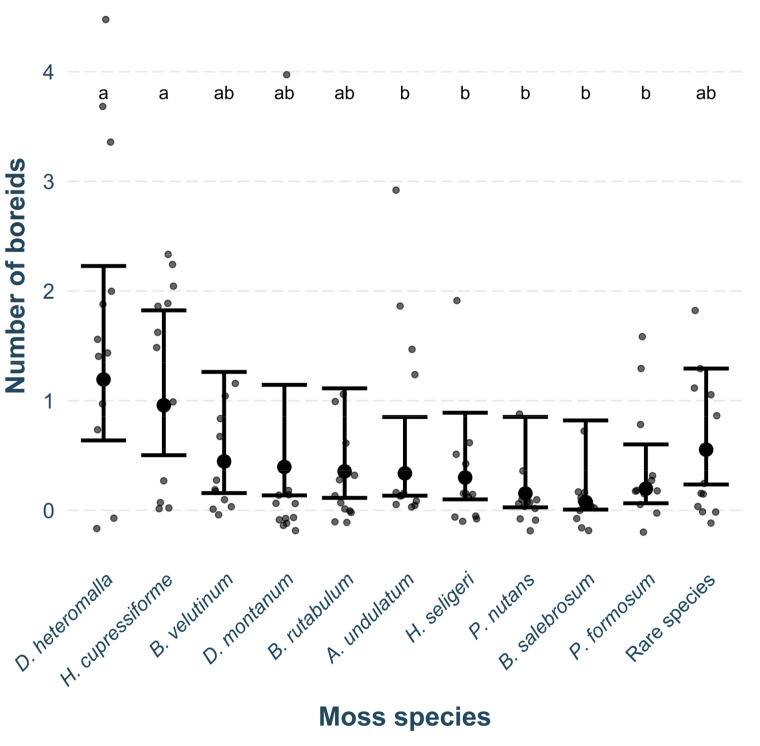
Model estimates for the number of boreids found on each moss species, assuming equal moss cover (mean ± 95% CI) presented alongside original data points representing raw observations. These estimates allow for a direct comparison of boreid presence across moss species while controlling for differences in moss cover. Different letters indicate moss species that differ significantly from each other.

**Figure 4 insects-15-00878-f004:**
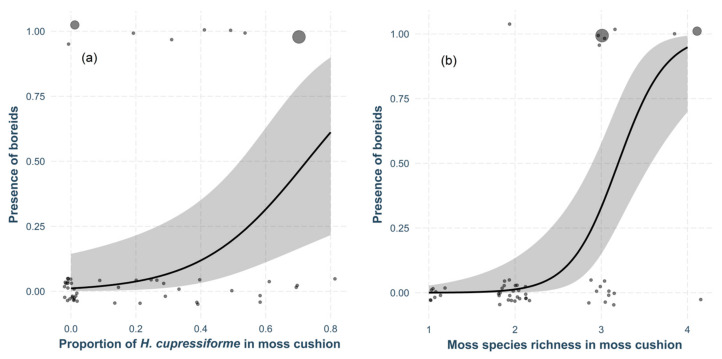
Model estimates for the probability of boreid presence based on (**a**) the proportion of *H. cupressiforme* in the moss cushion and (**b**) the species richness of mosses in the moss cushion (mean ± 95% CI) presented alongside original data points representing raw observations. The size of each point reflects its weight, corresponding to the number of boreids observed in each quadrat.

**Figure 5 insects-15-00878-f005:**
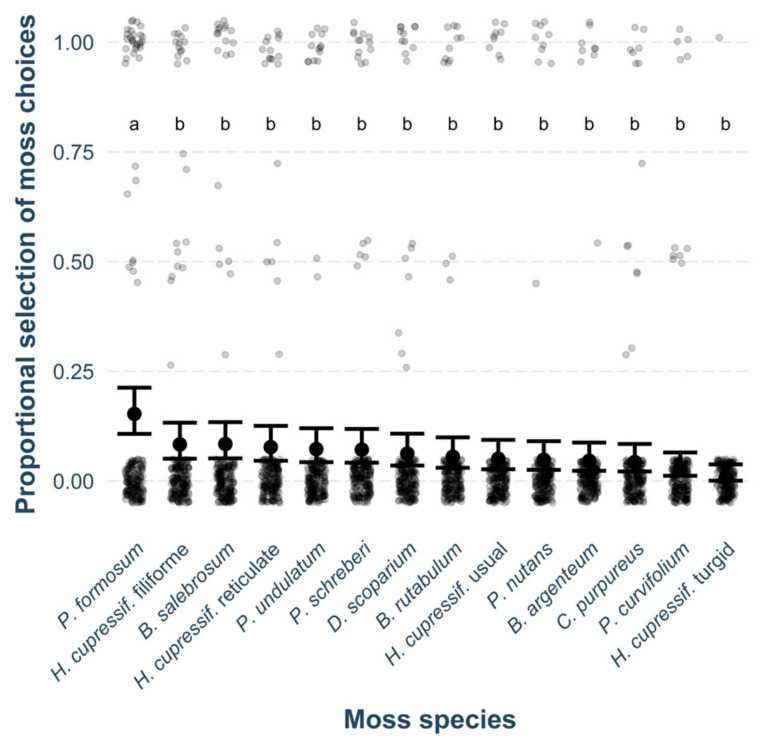
Model estimates for *B. hyemalis* selection tendencies among different moss species (mean ± 95% CI) presented alongside original data points representing raw observations. Different letters indicate moss species that differ significantly from each other.

## Data Availability

The raw data supporting the results of this article are available in the Appendix A provided with the manuscript.

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
