# Peer review of "Non-Random Distribution of *Boreus hyemalis* Among Bryophyte Hosts: Evidence from Field and Laboratory Tests"

_insects, 2024, doi:10.3390/insects15110878_

Round 1
Reviewer 1 Report
Comments and Suggestions for Authors
I enjoyed the paper by Pyszko et al and found the topic extremely interesting, particularly because the paper addresses some very rare insects about which little is known. The authors addressed the 'preferences' by B. hyemalis for moss species using field sampling and parallel laboratory choice tests. Overall, the authors demonstrated a mismatch that occurs between lab and field results and otherwise clear moss-associations for the insect in the field. The authors have not explained the mismatch, but, given the rarity of the species and the lack of research on bryophages in general, this result is certainly worthy of note.
I have a few minor suggestions for the authors that might improve the manuscript. Overall, I feel that the authors need to be more firm and consistent throughout the text, they seem to shift from an emphasis on host preferences to habitat/microhabitat preferences in different places - and indeed their laboratory choice experiment does not clarify the issue. The paper seems to really just indicate habitat preferences. Observations of longevity in the lab are largely irrelevant to indicating that the species might feed on mosses; because at low temperatures (8oC) specimens can often have very long lives.
ln 55 against from (delete one)
ln 68 missing closed bracket
ln 94-95 it is not clear what this means
ln 97 explain moss cushions
ln 100 not moss preference, but insect preference for mosses
methods general
- why was sampling only conducted in winter, some explanation of the species' seasonalities is merited
- how were specimens located in the first sampling, what was the sampling unit and how many samples were collected? This is also an issue in figure 1, we see several dots that represent sample results - but how was each dot calculated (sampling unit)
- is fagetum nudum in italics, please check
ln 154 sampled = collected
ln 166 repeated measures are indicated, but this does not appear in the analyses; also, the who choice experiment is difficult to understand, were the 16 observations regarded as new tests?
- including different morphotypes in a single experiment with different species gives rice to an unbalanced design with only one species having a nested factor. Perhaps to explain this, the best approach is to highlight that the test was a test of habitat preferences (although it confounds possible host preferences). The experiment should have been separated into 2. Some justification should be given.
Results
- expect F values for GLMs, why are these X2?
- Figure 2, nice graph, but x-axes are in different fonts
ln 243 and elsewhere - avoid most strongly, or most preferred, these are redundant terms, least preferred is fine
ln 264-265 - this contradicts the title and the introduction/objectives
ln 387-402 - this would be better placed in the introduction or under study species in the methods; it's the type of information that readers need upfront. The information also suggests that the work really only described potential habitat preferences and not host preferences (as in the title) - perhaps the title should be changed.
Because the species is rare - the authors might like to include an image, if possible.
-
Author Response
Dear Reviewers,
Thank you very much for your thorough and valuable feedback on our manuscript. We have carefully considered all your comments and suggestions, and we believe that the manuscript has improved as a result. Below are our detailed responses to each of your points.
Reviewer 1:
Comment:
I enjoyed the paper by Pyszko et al and found the topic extremely interesting, particularly because the paper addresses some very rare insects about which little is known. The authors addressed the 'preferences' by B. hyemalis for moss species using field sampling and parallel laboratory choice tests. Overall, the authors demonstrated a mismatch that occurs between lab and field results and otherwise clear moss-associations for the insect in the field. The authors have not explained the mismatch, but, given the rarity of the species and the lack of research on bryophages in general, this result is certainly worthy of note.
Response:
Dear reviewer, thank you for your positive and encouraging feedback. We are pleased that you found our paper interesting and appreciated its focus on rare insects with limited available research. We value all your comments and suggestions, which have hopefully helped to improve the manuscript.
Comment:
I have a few minor suggestions for the authors that might improve the manuscript. Overall, I feel that the authors need to be more firm and consistent throughout the text, they seem to shift from an emphasis on host preferences to habitat/microhabitat preferences in different places - and indeed their laboratory choice experiment does not clarify the issue. The paper seems to really just indicate habitat preferences. Observations of longevity in the lab are largely irrelevant to indicating that the species might feed on mosses; because at low temperatures (8oC) specimens can often have very long lives.
Response:
As you pointed out, there was a mismatch between field and laboratory results. In response to your observation and feedback from another reviewer, we have made several adjustments throughout the manuscript to better treat this discrepancy. We also addressed the inconsistency in emphasis between host preferences and habitat/microhabitat preferences by talking just about selection tendencies (especially in the laboratory test).
Regarding your concern about the irrelevance of longevity observations at low temperatures, we have removed this discussion from the manuscript, following a similar recommendation from another reviewer. However, I would like to clarify that, in fact, 8°C represents more of an upper threshold for survival for this particular winter species.
Comment:
ln 55 against from (delete one)
Response:
Thank you for catching this typo. We have corrected the sentence by deleting the extra word "against."
Comment:
ln 68 missing closed bracket
Response:
We have corrected the missing closed bracket, and upon further review, we found and fixed similar issues in a few other locations throughout the manuscript as well.
Comment:
ln 94-95 it is not clear what this means
Response:
We have revised the objectives in the Introduction section based on other suggestions, which has potentially resolved the ambiguity you mentioned. We hope the revised version is now clearer and better reflects our research aims.
Comment:
ln 97 explain moss cushions
Response:
Following other revisions to the objectives, this specific term has been removed. We hope this change addresses your concern and improves clarity.
Comment:
ln 100 not moss preference, but insect preference for mosses
Response:
Following other revisions to the objectives, this specific term has been removed. We hope this change addresses your concern and improves clarity
Comment:
why was sampling only conducted in winter, some explanation of the species' seasonalities is merited
Response:
The decision to conduct sampling only in winter was due to the specific seasonal activity pattern of B. hyemalis. This species is strictly a winter-active insect, with adults emerging and being active only during the colder months, exhibiting a unique phenology compared to most insects. Consequently, the only viable period for field observations and sampling is during the winter season, when individuals are actively present on or near mosses or snow. Outside this period, adults are no longer visible or available for sampling, making it impossible to study them. We have added this information to the Introduction section to provide a clearer understanding of the species' ecology and the rationale behind our sampling timeframe.
Comment:
How were specimens located in the first sampling, what was the sampling unit and how many samples were collected? This is also an issue in figure 1, we see several dots that represent sample results - but how was each dot calculated (sampling unit)
Response:
Specimens in the first phase of sampling were located by actively searching for adult B. hyemalis individuals directly on mosses within each site. A total of 72 individuals were sampled across six sites, with the number of individuals per site ranging from 10 to 15. Each moss species associated with a boreid within approximately 15 cm of the individual was recorded, and if multiple moss species were present, the association was estimated proportionally. Consequently, the sampling unit (i.e. response variable) in our study was defined at the moss species level, rather than at the level of individual boreids. After completing the field sampling, we identified 10 main moss species across the sites, with the remaining species grouped into a single category termed “rare species.” The response variable was the number of boreids found on each of these 11 categories within each locality. This number could be non-integer due to the proportional assignment of boreids when multiple moss species were within 15 cm of an individual (e.g., if one boreid was associated with two moss species in equal proportions, 0.5 boreids were assigned to each moss). Additionally, the response variable could take a value of zero if no boreids were found on a given moss species at a particular site. Unequal moss patch sizes were addressed by including moss cover as a covariate in the models.
We acknowledge that this sampling design is complex, but it was the only feasible solution that allowed us to account for the co-occurrence of multiple moss species in a relatively small area around each boreid and to compare the proportions of moss species found directly around boreids with those present at the site as a whole, in order to identify potential discrepancies and thus reveal any non-randomness in their distribution. We reflected this also in manuscript.
Comment:
is fagetum nudum in italics, please check
Response:
The term fagetum nudum has been checked and is formatted appropriately in italics, as per the conventions used for habitat types.
Comment:
ln 154 sampled = collected
Response:
Corrected
Comment:
ln 166 repeated measures are indicated, but this does not appear in the analyses; also, the who choice experiment is difficult to understand, were the 16 observations regarded as new tests?
Response:
In the laboratory preference test, each individual B. hyemalis was subjected to multiple observations (up to 16 measurements), but these were not treated as independent tests. Instead, the repeated measurements for each individual were accounted for by including the individual boreids as a random effect in a generalized linear mixed model (GLMM) – as already stated in manuscript. This approach allowed us to control for the non-independence of observations and appropriately model the variability within and between individual insects. To reduce potential biases from previous choices, the order of moss species in each trial was randomly changed, ensuring that any preference patterns were not influenced by the position of the mosses in the setup. We believe that the combination of using a mixed model structure with individuals as a random term and randomizing moss positions in each trial should sufficiently address the issue of repeated measures and independence. We have revised the text in the Methods section to clarify these details and improve the overall understanding of the experimental setup.
Comment:
Including different morphotypes in a single experiment with different species gives rice to an unbalanced design with only one species having a nested factor. Perhaps to explain this, the best approach is to highlight that the test was a test of habitat preferences (although it confounds possible host preferences). The experiment should have been separated into 2. Some justification should be given.
Response:
I acknowledge that including different morphotypes within a single species, alongside other moss species, may introduce an imbalance in the experimental setup. However, the primary reason for this was the notable morphological variability within Hypnum cupressiforme. We included the different morphotypes of H. cupressiforme in the same experiment to assess how within-species morphological differences might affect insect selection across moss species. I agree that separating the experiment into two—one testing inter-species selection tendencies and the other focused on morphotypes—would have offered a cleaner comparison. However, due to logistical constraints (the rarity of B. hyemalis and the difficulty in collecting enough specimens), we opted for a combined approach. Unfortunately, with the current data set, we cannot retrospectively separate the experiments as suggested. However, we commit to considering this approach in future studies to ensure a more balanced experimental design. In the meantime, we have made adjustments in other areas of the manuscript based on reviewer feedback to enhance clarity and rigor wherever possible.
Comment:
Expect F values for GLMs, why are these X2?
Response:
The X² values were reported because we used generalized linear mixed models and generalized linear models with binomial distribution. In GLMMs and binomial GLMS, the significance of model terms is typically tested using X²-tests on changes in deviance, rather than F-tests, due to the likelihood-based estimation approach and the incorporation of random effects. This method provides a more robust way of assessing the contribution of fixed effects when non-Gaussian distributions or random effects are involved.
Comment:
Figure 2, nice graph, but x-axes are in different fonts
Response:
Corrected
Comment:
ln 243 and elsewhere - avoid most strongly, or most preferred, these are redundant terms, least preferred is fine
Response:
Across the entire manuscript, we have removed redundant terms like "most strongly" and "most preferred," ensuring that only clearer and more concise phrasing is used where appropriate.
Comment:
ln 264-265 - this contradicts the title and the introduction/objectives
Response:
In response to this and after revising the objectives to better align with the goals of the study, we have updated the title to more accurately reflect the research focus. The new title, "Non-Random Distribution of Boreus hyemalis Among Bryophyte Hosts. Evidence from Field and Laboratory Tests", emphasizes the key aim of testing whether B. hyemalis shows a non-random distribution across moss species based on both field observations and controlled laboratory tests.
Comment:
ln 387-402 - this would be better placed in the introduction or under study species in the methods; it's the type of information that readers need upfront. The information also suggests that the work really only described potential habitat preferences and not host preferences (as in the title) - perhaps the title should be changed.
Response:
We have relocated the detailed information regarding the diet of Boreus hyemalis to the Introduction, where it provides context about the species' feeding habits and known associations with moss species. In the Discussion, we now reference these only briefly to avoid redundancy.
Moreover, in light of several revisions made throughout the manuscript, including those addressing objectives, analyses, and the overall focus of the study, we have indeed changed the title. The new title, now maybe accurately reflects the scope of the research and aligns more with the content and objectives presented in the manuscript.
Comment:
Because the species is rare - the authors might like to include an image, if possible.
Response:
We have added an image of Boreus hyemalis to the manuscript. We hope this addition will enhance the reader’s understanding and appreciation of the species.
Reviewer 2 Report
Comments and Suggestions for Authors
Overview
This reviewer identified several issues that undermine the conclusions of the study. The hypothesis is not clearly defined, and the assumption that H. cupressiforme is the preferred host of Boreus hyemalis is not adequately tested. Additionally, some of the results from the three approaches are contradictory, and the study lacks appropriate control groups to properly isolate moss preferences. The statistical methods employed are suboptimal, and the small sample size in laboratory tests, coupled with the absence of key ecological factors, further weakens the findings. Overall, the evidence and the speculative interpretations do not convincingly support the claim that Boreus hyemalis has a strong preference for H. cupressiforme. Rather, the results suggest that the insect may be an opportunistic generalist, without a specific preference for any particular moss species.
Introduction
The research question and hypothesis are not clearly formulated. The authors appear to have assumed that Hypnum cupressiforme is the preferred host for Boreus hyemalis without thoroughly testing this premise. The manuscript lacks a clear rationale for selecting H. cupressiforme among the many potential moss species.
The background provided is insufficient to establish the context for the study. The introduction should include more details on the ecological role of B. hyemalis and why H. cupressiforme was hypothesized to specifically choose over other moss species.
LL. 40–42: The background on the recent interest in host specificity of insect herbivores should be introduced using more directly relevant references than [1].
L. 43: The authors should replace [2] with a more directly relevant and recently published reference to better support the statement.
Methods
· There is a notable lack of appropriate control groups in both the field and laboratory studies. Without a proper control group to compare the results, it is difficult to properly interpret the results as the preference.
· The selection criteria for the 10 study sites are unclear, as are their ecological characteristics and the area of each site. Including a map would help clarify these details.
· L.117: The meaning and execution of the proportional estimate are unclear and should be elaborated for clarity.
· LL. 122-123: The methods in this section require further elaboration to improve clarity for readers.
· Section 2.2: The sample size is too small to draw reliable conclusions. Additionally, there is no mention of the age, mating experience, or sex ratio of the animals used in this test, nor any further tests to assess whether sex influenced moss preference.
· L.158: The rationale for selecting 8°C as the incubation temperature is unclear. Is this temperature high enough to induce hunger-based moss preference? What about its impact on their mating behavior and reproduction?
· L.160: The dimensions of the test arena should be specified.
· LL. 166-167: It is unclear how the animals were tested repeatedly and how pseudoreplication was avoided.
· LL. 168-175: The reason for testing only H. cupressiforme for four morphotypes is unclear. Could other moss species also exhibit different morphotypes, and if so, why were these not tested?
· LL. 207-210: The meaning of the "14 available positions" is unclear. What specific hypothesis was the permutation test designed to evaluate?
Results
· The figure captions are misleading. Figures are labeled as "model estimates," but they actually depict means with confidence intervals and raw data points. Clarification is needed regarding what the dots and circles in the figures represent.
· H. cupressiforme does not show a statistically significant preference over other moss species, yet the discussion implies otherwise. This discrepancy should be addressed.
· In Figure 2, it would make more logical sense to switch the order of the a) and b) panels.
· In Figure 3, only P. formosum exhibits a statistically significant preference. However, the discussion continues to assert a significant preference for H. cupressiforme, which is not supported by the results.
· In Section 3.2, the sample size is omitted and should be provided.
· LL. 249-251: The mention of the insects’ longevity in captivity seems unrelated to the main focus of the study.
· LL. 255-260: The rationale for generating these orders is unclear. What was the intended purpose?
Discussion
· The findings from the three approaches are inconsistent with one another, and the manuscript does not adequately address or reconcile these contradictions. The discussion lacks sufficient evidence to support the conclusion that H. cupressiforme is the preferred moss, relying heavily on speculation. As a result, the discussion fails to properly address the inconsistencies between the field and laboratory results, with the proposed mechanisms for moss preference not backed by solid data. This reviewer recommends that the authors refine the arguments presented in the Discussion and prioritize designing a more robust study in the future to address these gaps.
· Statements such as "Unfortunately, we did not include D. heteromalla in the lab test" (L336) and "This could have provided an explanation" (L337) are speculative and weaken the scientific value of the study and do not offer meaningful insights.
· LL. 339-341: The authors acknowledge that their field observation and data collection methods were not systematic or thorough, which undermines the reliability of the findings.
· LL. 362-365: If the morphological characteristics mentioned are important regarding the insects’ preference, they should have been included in the Methods and Results section, not introduced for the first time in the Discussion.
· LL. 369-373: If the authors wish to compare their findings with those of the beetle family, more elaboration and detailed comparison are needed.
· LL. 374-375: The prediction that unfamiliar moss would be avoided contradicts the authors' following proposition that the insect might feed on diverse moss species to neutralize potential toxic metabolites. A more careful and consistent argument is needed.
· LL. 382-386: This sentence suggests a correlation rather than a specific intentional preference, and this should be clarified.
· LL. 387-392: The claim that the diet of Boreus is still under debate is contradictory, as multiple studies have already well established its bryophagy. Without a solid biological basis or clear functional relevance of H. cupressiforme to Boreus, the overarching hypothesis does not hold.
· LL. 404-414: This paragraph seems out of context and does not serve a clear purpose within the section.
Conclusion
· The conclusion that B. hyemalis has a strong preference for H. cupressiforme is not adequately supported by the data.
· LL. 416-418: This reviewer did not find sufficient evidence to support this conclusion.
· LL. 419-421: This specific claim was not directly tested in the present study.
Minor points
There are several formatting errors and inconsistencies throughout the text. Below are some examples noted by this reviewer, though this list is not exhaustive:
· L. 59: "Wheeler, 2003" should be [13].
· L. 221 vs. L. 222: Ensure consistency in the presentation of statistical data.
· L. 225 vs. L. 232: Ensure consistency in formatting the P values.
The references also contain various minor editorial errors. Some examples include:
· LL. 451-453
· L. 478
· L. 493
· L. 511
· LL. 518-522
Comments on the Quality of English LanguageThere are several instances where the English is unclear due to issues with sentence structure, grammar, or lack of flow. The following sections require clarification and refinement:
· LL. 134-136: Requires improvement in sentence structure and clarity.
· LL. 193-195: The wording and flow need revision for better readability.
· LL. 382-386: The phrasing is unclear and should be reworked for clarity.
· L. 392: The term "theory" is misused and should be corrected.
Author Response
Dear Reviewers,
Thank you very much for your thorough and valuable feedback on our manuscript. We have carefully considered all your comments and suggestions, and we believe that the manuscript has improved as a result. Below are our detailed responses to each of your points.
Reviewer 2:
Comment:
This reviewer identified several issues that undermine the conclusions of the study. The hypothesis is not clearly defined, and the assumption that H. cupressiforme is the preferred host of Boreus hyemalis is not adequately tested. Additionally, some of the results from the three approaches are contradictory, and the study lacks appropriate control groups to properly isolate moss preferences. The statistical methods employed are suboptimal, and the small sample size in laboratory tests, coupled with the absence of key ecological factors, further weakens the findings. Overall, the evidence and the speculative interpretations do not convincingly support the claim that Boreus hyemalis has a strong preference for H. cupressiforme. Rather, the results suggest that the insect may be an opportunistic generalist, without a specific preference for any particular moss species.
Response:
Dear Reviewer, thank you for your comprehensive feedback and for highlighting multiple aspects of the manuscript that could be improved. We have carefully reviewed all your comments and have made revisions and clarifications throughout the manuscript in response to your concerns. Below, we provide detailed responses to each of your points. Regarding the statistical analysis, we acknowledge your concern about the suboptimal nature of some methods. However, your feedback did not specify particular issues with the statistical approach, apart from concerns about pseudoreplication, which we have addressed in detail in the respective sections. We believe that the use of mixed models with carefully selected structure and distributions is a robust approach for analyzing our data. We hope that the revisions we made and the clarifications provided in the following responses address your concerns adequately.
Comment:
The research question and hypothesis are not clearly formulated. The authors appear to have assumed that Hypnum cupressiforme is the preferred host for Boreus hyemalis without thoroughly testing this premise. The manuscript lacks a clear rationale for selecting H. cupressiforme among the many potential moss species.
Response:
We appreciate your concern and would like to clarify our approach. At the start of our study, we did not assume that Hypnum cupressiforme was the preferred host for Boreus hyemalis. Our primary objective was to test whether Boreus hyemalis showed non-random selection for any moss species in its habitat, or if the distribution was random. The selection of H. cupressiforme for more detailed laboratory testing was not based on an a priori hypothesis about its preference but rather on the fact that H. cupressiforme is morphologically diverse and is one of the most abundant moss species in the habitats where B. hyemalis was frequently observed. This allowed us to explore potential inter-morphotype variations in host preference. In response to your suggestion, we have revised the Introduction and Materials and Methods section to make our research question and hypothesis clearer.
Comment:
The background provided is insufficient to establish the context for the study. The introduction should include more details on the ecological role of B. hyemalis and why H. cupressiforme was hypothesized to specifically choose over other moss species.
Response:
We agree with your suggestion to include more details on the ecological role of B. hyemalis. In response, we have expanded the Introduction to provide additional information on the species' ecological characteristics. This provides a clearer context for understanding its interactions with mosses. However, as mentioned in a previous response, we did not initially hypothesize that H. cupressiforme would be specifically preferred over other moss species. Our study aimed to investigate the broader question of whether B. hyemalis exhibits any non-random association with specific moss species in the field. The focus on H. cupressiforme and its morphotypes in the laboratory experiments was based on its abundance and morphological variability, which allowed us to test potential differences in insect responses. This morphological variability is introduced in M & M section now.
Comment:
- 40–42: The background on the recent interest in host specificity of insect herbivores should be introduced using more directly relevant references than [1].
Response:
We have completely revised the first sentence of the introduction to better align with both the rest of the paragraph and the overall introduction. This change also addresses the need for more relevant references regarding host specificity in insect herbivores.
Comment:
- 43: The authors should replace [2] with a more directly relevant and recently published reference to better support the statement.
Response:
We have retained the sentence and supported it with a more directly relevant and recently published reference.
Comment:
There is a notable lack of appropriate control groups in both the field and laboratory studies. Without a proper control group to compare the results, it is difficult to properly interpret the results as the preference.
Response:
We acknowledge that an explicit control group was not included in the laboratory experiments; however, this limitation is partially compensated by the large number of moss species tested. In this context, rather than discussing "preferences," we have adjusted our terminology to refer to "selection tendencies," and we have revised the manuscript accordingly. Our primary results and conclusions are based on the field tests, with the laboratory test included mainly for completeness, serving to support and complement the field observations. Regarding the field experiments, we respectfully believe that a traditional control group is not applicable here. The field observations are based on comparing the number of boreids found on different moss species relative to their coverage. The implicit control here is the expected random distribution of boreids if they did not prefer any particular moss. Deviations from this expected distribution indicate some none-randomness in selection. We have emphasized this point in the throughout the manuscript including the title based on your comment.
Comment:
The selection criteria for the 10 study sites are unclear, as are their ecological characteristics and the area of each site. Including a map would help clarify these details.
Response:
We have now clarified the selection criteria for the study sites, which were chosen based on their suitability as habitats for B. hyemalis, specifically targeting locations with appropriate moss cover. These criteria were applied to ensure that each site had a high potential for yielding boreids, which was essential for our field experiments. We also created a map illustrating all visited locations, highlighting the sites with sufficient boreid numbers (marked in blue), those where boreids were found in low numbers (red), and those used for the laboratory tests (yellow).
Comment:
L.117: The meaning and execution of the proportional estimate are unclear and should be elaborated for clarity.
Response:
We agree that the description of the proportional estimate relies too much on the reference and needed further clarification. We have revised it to provide a more detailed explanation of how we conducted this estimation (with example).
Comment:
- 122-123: The methods in this section require further elaboration to improve clarity for readers.
Response:
We agree that additional clarification was necessary. We have revised the Methods section to provide a more detailed explanation of how this was done
Comment:
Section 2.2: The sample size is too small to draw reliable conclusions. Additionally, there is no mention of the age, mating experience, or sex ratio of the animals used in this test, nor any further tests to assess whether sex influenced moss preference.
Response:
Boreus hyemalis is a rare and difficult-to-find species. Collecting individuals for laboratory experiments required substantial effort due to their low abundance, cryptic nature and low tolerance to standard laboratory temperatures. Despite these challenges, we were able to sample 60 individuals used approx. half of them in initial laboratory tests (30 were preserved in ethanol), with a final dataset comprising 22 individuals that completed up to 16 repeated measurements each. Given the rarity of the species and the intensive effort required to obtain specimens, we believe that our sample size is appropriate and sufficient to draw reliable conclusions within the context of our study. As the boreids were collected from the field, it was not feasible to determine their exact age or mating experience. We have added the information to the Methods section to clarify this point. Each boreid was housed individually during the experiments, and their sex was documented and included as an explanatory variable in our analyses. Therefore, the sex ratio is inherent in the dataset, and the influence of sex on moss preference was thoroughly assessed. As reported in the Results section, no significant differences in moss preferences between sexes were detected (both P > 0.050), although males showed a slight, non-significant selection tendency for P. formosum. This observation is discussed in the Discussion section.
Comment:
L.158: The rationale for selecting 8°C as the incubation temperature is unclear. Is this temperature high enough to induce hunger-based moss preference? What about its impact on their mating behavior and reproduction?
Response:
We have revised the Methods section to provide a detailed explanation. Shortly, the choice of 8 °C was a compromise between two conflicting requirements. On one hand, we needed to keep the individuals alive as long as possible, which, given the biology of Boreus hyemalis. On the other hand, the growth chamber we used had technical limitations that allowed a minimum temperature of 8 °C for extended operation. Therefore, 8 °C was the lowest temperature we could reliably maintain for the duration of the experiments, without destroying the engine. While we acknowledge that a higher and thus suboptimal temperature might have increased their metabolic rate and led to earlier onset of hunger, all individuals were subjected to the same conditions. We believe this consistency minimizes any bias in the comparative results among different moss species. As for the impact on mating behavior and reproduction, all individuals were kept separately from the time of collection, through acclimation, and during the experiments until their death. This was done to minimize pseudoreplication and ensure that individual behaviors could be accurately recorded. Therefore, mating behavior and reproduction were not factors that could have influenced the results of our study.
Comment:
L.160: The dimensions of the test arena should be specified.
Response:
We included necessary details about arenas.
Comment:
- 166-167: It is unclear how the animals were tested repeatedly and how pseudoreplication was avoided.
Response:
We agree that further clarification was needed regarding the repeated testing of animals and how pseudoreplication was addressed. In our experiments, each boreid individual was tested separately to prevent any interactions between individuals that could influence their behavior. Each individual was placed alone in a testing arena. This setup ensured that the observed behaviors were solely attributable to the individual's preferences without social influences. Each boreid underwent up to 16 repeated trials over the course of the study, with adequate intervals between trials to reduce any carry-over effects from previous exposures. By recording the identity of each individual and treating individual boreids as a random effect in our generalized linear mixed models, we accounted for the repeated measures on the same individuals to control for within-individual correlations and avoid pseudoreplication. We have revised the Methods section to provide a more detailed explanation of our experimental design and statistical analysis.
Comment:
- 168-175: The reason for testing only H. cupressiforme for four morphotypes is unclear. Could other moss species also exhibit different morphotypes, and if so, why were these not tested?
Response:
- cupressiforme is well-known for its significant morphological variability, which is greater than that observed in most other moss species. This morphotypes differ in their physical structure and potentially in their ecological interactions. Our decision to test these four morphotypes was based because of our previous research, where we explored the ecological significance of H. cupressiforme morphotypes. We are also planning future studies focusing on this species, including metabolomic analyses, to further understand the biochemical and physiological differences among its morphotypes. Thus, while other moss species may exhibit some morphological variation, H. cupressiforme's variability is particularly pronounced and relevant to our research objectives. We have updated the manuscript to include this explanation in the Materials and Methods section to clarify our rationale.
Comment:
- 207-210: The meaning of the "14 available positions" is unclear. What specific hypothesis was the permutation test designed to evaluate?
Response:
The term "14 available positions" refers to the total number of positions in rank that the four selected moss species could occupy within our experimental setup. This framework was established to assess whether the observed ordering of these moss species deviates significantly from what would be expected by random chance. The permutation tests were conducted post hoc to evaluate whether the observed rankings of the selected moss species were significantly different from random arrangements. The permutation tests were performed after observing the results, serving as a supplementary analysis to support our discussion. This approach was intended to provide probabilistic evidence that certain moss species are indeed less preferred by boreids beyond what would be expected from a random distribution. We ensured that this post hoc analysis did not involve generating hypotheses after knowing the results (HARKing), as it was solely used to enhance the interpretative strength of our findings. We have revised the manuscript to include a more detailed explanation of the "14 available positions" and the rationale behind conducting the permutation test.
Comment:
The figure captions are misleading. Figures are labeled as "model estimates," but they actually depict means with confidence intervals and raw data points. Clarification is needed regarding what the dots and circles in the figures represent.
Response:
To enhance clarity and accurately represent the data, we have revised the figure captions to explicitly indicate that the figures display both model estimates (means ± 95% confidence intervals) and raw data points.
Comment:
- cupressiforme does not show a statistically significant preference over other moss species, yet the discussion implies otherwise. This discrepancy should be addressed
Response:
You are absolutely right regarding the laboratory tests. Additionally, the inclusion of all four morphotypes of H. cupressiforme showed a highly varied ranking of these morphotypes, which further highlights the lack of a consistent preference for H. cupressiforme as a whole in the laboratory setting. We have adjusted the discussion to reflect this and have moderated our statements accordingly. However, in the field studies, H. cupressiforme was indeed one of the mosses with the highest number of boreids, along with Dicranella heteromalla, and showed a tendency to be selected more frequently than other moss species. Moreover, in the second field experiment, the presence of Hypnum was the most significant explanatory factor for boreid presence. We have made these distinctions clearer in the discussion to avoid any confusion between the field and laboratory results.
Comment:
In Figure 2, it would make more logical sense to switch the order of the a) and b) panels.
Response:
We appreciate your suggestion to rearrange the panels in Figure 2 for improved logical flow. However, we have intentionally maintained the current order of panels a) and b) to reflect the significance and the sequence in which these explanatory variables were incorporated into our statistical model. Specifically, the proportion of H. cupressiforme in the moss cushion was identified as the most significant predictor of boreid presence and was thus presented first. The species richness of mosses followed in order of their significance and inclusion into the model.
Comment:
In Figure 3, only P. formosum exhibits a statistically significant preference. However, the discussion continues to assert a significant preference for H. cupressiforme, which is not supported by the results.
Response:
We have addressed this concern in our response to one of your previous comment, where we clarified that H. cupressiforme did not show a consistent preference in the laboratory experiments and that our discussion has been adjusted to reflect this fact. We now clearly differentiate between the findings in the field, where H. cupressiforme was frequently observed, and the laboratory results, where it did not exhibit a statistically significant preference.
Comment:
In Section 3.2, the sample size is omitted and should be provided.
Response:
We agree that specifying the sample size is essential for the clarity and reproducibility of our study. To address this, we have added the sentence.
Comment:
- 249-251: The mention of the insects’ longevity in captivity seems unrelated to the main focus of the study.
Response:
We have removed the sentence from the Results section. Instead, we now provide this information in the Materials and Methods section, where it is integrated to discuss the characteristics of the boreids, thereby also addressing one of your previous comments regarding these aspects.
Comment:
- 255-260: The rationale for generating these orders is unclear. What was the intended purpose?
Response:
The primary purpose of this analysis was to provide probabilistic evidence supporting our discussion that certain moss species are less preferred by boreids than others. This approach allowed us to move beyond descriptive observations and statistically validate the non-random distribution of boreid preferences. We acknowledge that this permutation test was performed after observing the results, serving as a supplementary analysis to reinforce our findings without introducing new hypotheses post hoc (HARKing). This methodology ensures that our interpretations are grounded in both observed data and statistical validation. To enhance the clarity of our manuscript, we have revised the relevant sections to explicitly detail the purpose and rationale of the permutation test, as outlined in the corrections above.
Comment:
The findings from the three approaches are inconsistent with one another, and the manuscript does not adequately address or reconcile these contradictions. The discussion lacks sufficient evidence to support the conclusion that H. cupressiforme is the preferred moss, relying heavily on speculation. As a result, the discussion fails to properly address the inconsistencies between the field and laboratory results, with the proposed mechanisms for moss preference not backed by solid data. This reviewer recommends that the authors refine the arguments presented in the Discussion and prioritize designing a more robust study in the future to address these gaps.
Response:
We agree that our discussion required refinement to better address the discrepancies between the field and laboratory findings and to provide a more balanced interpretation of the results. To address this, we: 1) Restructured the discussion to clearly delineate the insights provided by each experimental approach and to highlight their respective limitations; 2) Moderated our language regarding the preference for H. cupressiforme, ensuring that it is presented as a potential preference in field observations, and as selection tendencies in lab observations. We now explicitly state that while H. cupressiforme was frequently observed in the field, it did not show a statistically significant preference in the laboratory tests. 3) Added a section in the conclusion that discusses the need for a more robust study design to better understand moss preferences in Boreus hyemalis. This includes some suggestions. We believe these revisions provide a more accurate interpretation of our findings and better acknowledge the limitations of our study.
Comment:
Statements such as "Unfortunately, we did not include D. heteromalla in the lab test" (L336) and "This could have provided an explanation" (L337) are speculative and weaken the scientific value of the study and do not offer meaningful insights.
Response:
We understand that the original wording appeared speculative and undermined the clarity of our conclusions. We have revised these statements to remove language that could be perceived as weakening the scientific rigor of the manuscript, while still clearly acknowledging the limitations of the study.
Comment:
- 339-341: The authors acknowledge that their field observation and data collection methods were not systematic or thorough, which undermines the reliability of the findings.
Response:
We have addressed this concern in our response to your previous comment, where we revised the language to avoid any speculative phrasing that might undermine the scientific value of the study. The updated text now presents the limitations objectively, without suggesting a lack of systematic approach in our methodology.
Comment:
- 362-365: If the morphological characteristics mentioned are important regarding the insects’ preference, they should have been included in the Methods and Results section, not introduced for the first time in the Discussion.
Response:
We agree with your suggestion and have revised the manuscript accordingly. The detailed descriptions of the H. cupressiforme morphotypes are now presented in the Methods section, where we outline the morphological traits used to define each morphotype and explain their selection. In the Discussion section, we have shortened the relevant paragraph to focus only on the results and implications, without reintroducing the detailed morphotype descriptions.
Comment:
- 369-373: If the authors wish to compare their findings with those of the beetle family, more elaboration and detailed comparison are needed.
Response:
We appreciate your suggestion to provide a more detailed comparison between our findings and those reported for beetles of the family Byrrhidae. However, our intention was not to conduct a direct comparison between boreid preferences and beetle behaviors, as the experimental design and ecological contexts of the studies differ significantly. The reference to Byrrhidae beetles was used merely as an illustrative example to highlight that differences in preferences toward various morphotypes can also occur in other bryophagous insects, even when morphological variations are minimized.
Comment:
- 374-375: The prediction that unfamiliar moss would be avoided contradicts the authors' following proposition that the insect might feed on diverse moss species to neutralize potential toxic metabolites. A more careful and consistent argument is needed.
Response:
We understand your concern regarding the potential contradiction. However, we believe that these two ideas can coexist within the framework of the dietary mixing hypothesis. It is plausible that boreids actively select from a range of familiar moss species to balance their diet and neutralize any potential toxic effects, while simultaneously avoiding unfamiliar moss species due to the perceived risk of unknown. This behavior is consistent with that observed in other generalist herbivores, where individuals seek dietary diversity but are cautious of novel food sources. We respectfully suggest keeping the current interpretation, as it aligns with the broader theory of dietary mixing. However, we revise the discussion slightly to better articulate this idea and avoid any perceived contradictions.
Comment:
- 382-386: This sentence suggests a correlation rather than a specific intentional preference, and this should be clarified.
Response:
We appreciate your feedback and agree that the original phrasing might have suggested a stronger inference about intentional preferences than warranted by the data. To address this, we have rephrased the relevant sentences in the Discussion to clarify that the observed association between boreids and certain moss species may represent a correlation influenced by environmental availability, rather than a definitive preference by the insects. We hope this revision addresses your concern
Comment:
- 387-392: The claim that the diet of Boreus is still under debate is contradictory, as multiple studies have already well established its bryophagy. Without a solid biological basis or clear functional relevance of H. cupressiforme to Boreus, the overarching hypothesis does not hold.
Response:
We agree that the question of Boreus diet has been resolved, with multiple studies confirming its bryophagous nature. To reflect this consensus, we have revised the relevant text to clarify that this was previously a debated topic, but the current understanding firmly establishes Boreus as a phytophagous insect feeding predominantly on mosses. Additionally, we have removed the paragraph to Introduction section and note that while some studies suggest an association between Boreus and specific mosses like Hypnum cupressiforme, no critical evaluation has definitively established a non-randomness beyond what might be expected based on their field abundance. We hope this addresses your concerns about the biological relevance of our hypothesis.
Comment:
- 404-414: This paragraph seems out of context and does not serve a clear purpose within the section.
Response:
We agree with your assessment that this paragraph was out of context and did not contribute effectively to the discussion. As a result, we have deleted this portion from the Discussion section. We retained only a brief mention of one key aspect and moved it to the Methods section to maintain the necessary information while improving the overall flow of the manuscript.
Comment:
The conclusion that B. hyemalis has a strong preference for H. cupressiforme is not adequately supported by the data.
Response:
As mentioned in one of our previous response, we have revised the conclusion to better reflect the nuances of our findings. Specifically, we have moderated our statements regarding H. cupressiforme preference, acknowledging the discrepancies between the field and laboratory results. While H. cupressiforme was frequently observed in the field, it did not show a statistically significant preference in the laboratory experiments. We have updated the conclusion to emphasize this point, stating that the non-randomness of selection for H. cupressiforme observed in the field should not be overinterpreted, given the lack of significant support from laboratory data. The revised conclusion now highlights the need for further research to better understand these patterns.
Comment:
- 416-418: This reviewer did not find sufficient evidence to support this conclusion.
Response:
We have completely rewritten the Conclusion to better align it with the results and to address the concerns you raised. We hope that the revised Conclusion resolves this issue (and thus also your comment) and provides a clearer interpretation of our findings.
Comment:
- 419-421: This specific claim was not directly tested in the present study.
Response:
We have completely rewritten the Conclusion to better align it with the results and to address the concerns you raised. We hope that the revised Conclusion resolves this issue (and thus also your comment) and provides a clearer interpretation of our findings.
Comment:
There are several formatting errors and inconsistencies throughout the text. Below are some examples noted by this reviewer, though this list is not exhaustive:
- 59: "Wheeler, 2003" should be [13].
Response:
Corrected
Comment:
- 221 vs. L. 222: Ensure consistency in the presentation of statistical data.
Response:
Corrected
Comment:
- 225 vs. L. 232: Ensure consistency in formatting the P values.
Response:
Corrected
Comment:
The references also contain various minor editorial errors. Some examples include:
- 451-453
- 478
- 493
- 511
- 518-522
Response:
Corrected
Comment:
Comments on the Quality of English Language
There are several instances where the English is unclear due to issues with sentence structure, grammar, or lack of flow. The following sections require clarification and refinement:
- 134-136: Requires improvement in sentence structure and clarity.
Response:
We have revised the sentence (and whole paragraph) to improve structure and clarity. We focus on enhancing readability and ensuring that the flow of information is smoother.
Comment:
- 193-195: The wording and flow need revision for better readability.
Response:
The revisions focus on enhancing readability and ensuring that the description of the statistical model is clearer.
Comment:
- 382-386: The phrasing is unclear and should be reworked for clarity.
Response:
Revised, see one of the previous comments.
Comment:
- 392: The term "theory" is misused and should be corrected.
Response:
We have revised the sentence to replace the term "theory" with the more appropriate term.
Round 2
Reviewer 2 Report
Comments and Suggestions for Authors
Thank you for your thorough revision of the manuscript in response to this reviewer’s extensive comments. The changes you made addressed most of the points and have significantly improved the overall quality of the manuscript.